# Medical Students’ Knowledge and Attitudes Regarding Justice-Involved Health

**DOI:** 10.3390/healthcare9101302

**Published:** 2021-09-30

**Authors:** Margaret English, Fatimata Sanogo, Rebecca Trotzky-Sirr, Todd Schneberk, Melissa Lee Wilson, Jeffrey Riddell

**Affiliations:** 1Keck School of Medicine, University of Southern California, 1975 Zonal Avenue, Los Angeles, CA 90033, USA; 2Department of Preventive Medicine, Keck School of Medicine, University of Southern California, 2001 N. Soto Street, Los Angeles, CA 90033, USA; fsanogo@usc.edu (F.S.); melisslw@usc.edu (M.L.W.); 3Department of Emergency Medicine, LAC + USC, 1200 N State Street Rm 1011, Los Angeles, CA 90033, USA; RTrotzky-Sirr@dhs.lacounty.gov (R.T.-S.); tschneberk@gmail.com (T.S.); jeffriddell@gmail.com (J.R.)

**Keywords:** curriculum, ethics, attitude of health personnel, education, medical, medicine, correctional facilities, social justice

## Abstract

Despite the demonstrated need for sustainable and effective carceral health care, justice-involved medical education curricula are limited, and it’s unclear if informal clinical education is sufficient. Investigators aimed to quantify medical student involvement with carceral populations and explore how students’ knowledge of and attitudes towards justice-involved patients changed over the course of their training. A survey was designed by the investigators and sent to all current medical students at a single United States medical school. Stata 14.0 was used to compare results between the years of medical school. Differences between groups were tested using linear regression. Most 4th year students reported working in a carceral health setting. An increase in overall knowledge of justice-involved patients was observed as carceral medicine education (*p*_trend_ = 0.02), hours worked in a jail (*p*_trend_ < 0.01), and substance abuse training (*p*_trend_ < 0.01) increased. Overall attitude score increased with the students’ reported number of hours working in a jail (*p*_trend_ < 0.01) and the amount of substance abuse training (*p*_trend_ < 0.01). Finally, we found a trend of increasing knowledge and attitude scores as the year of standing increased (*p*_trend_ < 0.01). Our data suggest that most USC medical students work in a carceral setting during medical school. Didactic and experiential learning opportunities correlated with improved knowledge of and attitude toward justice-involved patients, with increases in both metrics increasing as the year in medical school increased. However, senior medical students still scored poorly. These findings underscore the need for a formal curriculum to train our healthcare workforce in health equity for carceral populations.

## 1. Introduction

Justice-involved individuals (defined as people who are recently or currently incarcerated or on parole) have unique health concerns that are not being addressed by the current US healthcare system and physician workforce [1]. Chronic diseases, mental health problems, substance use disorders, and infectious diseases are exacerbated by access barriers to health services both in the community and while justice-involved [2]. Despite the need for sustainable and effective carceral health care, recruiting well-trained doctors to work in correctional settings is difficult: there is not a sufficient supply of properly trained healthcare professionals to serve this uniquely vulnerable population. These difficulties may stem from widespread concerns about provider safety in caring for carceral populations as well as lack of prestige of correctional health jobs [3].

Much of the existing literature around justice-involved health focuses on the need for partnership between Academic Medicine and Correctional Facilities [4,5,6]. In this mutually beneficial partnership, jails and prisons allow students to see the value of providing meaningful care to the disadvantaged while educating developing clinicians on unique pathologies and the critical importance of primary and chronic disease management. Academic centers can offer expertise in evidence-based practice and education regarding structural factors contributing to health disparities in the justice-involved population [4]. Given this interdependence, medical coursework should provide both didactic training and exposure to the specific health needs of this population as part of a holistic medical education [5]. One study, which performed extensive focus group interviews with carceral health care providers, suggested that a comprehensive curricula for medical learners could include six main subject areas: the demographic characteristics of the population, common conditions which require clinical expertise, public health opportunities, ethical considerations when treating incarcerated patients, medical-legal issues, and practical knowledge of the structure and administration of the correctional health system [7].

Despite this imperative, there are few published medical school curricula for justice-involved populations. A variety of correctional health opportunities do exist, but most are institution-specific and overwhelmingly limited to optional electives [3,8,9,10]. One study found 17 discrete programs with correctional healthcare opportunities for medical students. Of these, several primarily trained residents, and most gave no mention of formal instruction in ethics, vulnerable populations, or structural causes of illness [10]. At one large Texas institution where nearly all medical students contribute to the care of incarcerated patients, authors discussed the difficulties that medical learners face when they are exposed to the harsh realities of care delivery behind bars without a curricular framework in which to scaffold their experiences [11].

While these opportunities often contribute to social awareness, positive feelings, and subjectively improved academic competence for participants, their small sample sizes, reliance on qualitative analysis, and self-selection of students participating in the given electives limits their generalizability. There is a paucity of studies that detail medical students’ experiences with justice-involved healthcare in the pre-clinical years or during the required core rotations of their clinical years. As such, a broader view of students’ involvement with this population remains cloudy.

Therefore, the aim of this study was to determine the extent of medical student involvement with the justice-involved population at a large urban medical school and explore students’ knowledge of and attitudes towards justice-involved patients. Specifically, we aimed to determine (1) if medical students are exposed to the justice-involved population while in medical school, and (2) how individual demographics, experiences, and year in school relate to knowledge of and attitudes towards the justice-involved population.

## 2. Materials and Methods

### 2.1. Study Design and Setting

We performed a cross-sectional survey of medical students at a single United States (US) medical school. A survey was chosen as the data collection method because we wanted to measure metrics (attitudes, beliefs) that are not directly observable. The study took place at the Keck School of Medicine of the University of Southern California (USC), at which the majority of clinical rotations occur in the Los Angeles County + USC (LAC + USC) Hospital. LAC + USC in the main health care site for the Twin Towers Correctional Facility, which is the nation’s largest jail and mental health facility. The survey was conducted in October and November 2019.

While the Keck School curriculum has no formal carceral-health requirement, students participate in clinical rotations on which they participate in the care of justice-involved patients. These experiences are largely arbitrary and are not consistent across rotations. The informal curriculum may include bedside teaching while providing care to inmates while on different surgical and medical services in the hospital, but these experiences are largely unstructured. Students do not self-select to participate in these experiences, and it is possible that every medical student could participate in the care of a justice-involved patient by graduation.

### 2.2. Selection of Participants

Participants were current USC medical students. Students in other programs affiliated with the medical school (Master’s degree only) were ineligible to participate.

Recruitment of study subjects was performed via institutional email—a short description of the study was emailed to all current students from years 1–4, including those in dual degree years (MD-PhD, MD-MPH, MD-MBA). Two weeks into the study period, reminder emails were sent to those students who had not yet completed the survey. Emails included a Qualtrics survey link which took participants directly to the survey. Each student was allowed to take the survey once. While participation was optional, the first 375 participants who completed the survey were given an $8 Amazon gift card as an incentive.

### 2.3. Protocol

After a detailed literature review and interviews with correctional health providers at two institutions (LAC + USC and the Twin Towers Correctional Facility), we developed a brief questionnaire modeled after the existing Substance Abuse Attitudes Survey (SAAS) [12] to expand its relevance to carceral populations. The SAAS is used to measure the attitudes of clinicians and medical students towards substance use. It has been used most notably in undergraduate medical education to measure changes in attitudes following educational experiences. The SAAS is relevant because justice-involved people and substance use disorders are closely linked—approximately half of the US carceral population meet diagnostic criteria for abuse or dependence [13], and there is pervasive stigma at the intersection of incarceration and substance use [14]. Authors referenced an institutional (LAC + USC) Emergency Medicine Jail Medicine Guide to develop the final survey to include 3 domains: exposure, knowledge, and attitudes. After development, the survey was sent to the Medical Education Council (MEC) at the Keck School of Medicine for evaluation by those with significant expertise in survey creation and administration. The authors and the council iteratively revised the survey items for clarity and relevance, and it was pilot tested with 3 Keck faculty members. Students were not used for pilot testing because this would have excluded them from participating in the study. Pilot testers read the study items out loud to themselves and repeated back to investigators how they interpreted each item. No significant changes were made after pilot testing. We designed the survey to be completed in less than 10 min. This survey was approved in English only and was not translated to other languages. The study was approved by the USC Health Science Campus IRB.

The survey included demographic questions followed by questions about personal exposure to justice-involved patients (“how many hours have you worked in the jail ward? How many hours of formal education have you received?”). Fifteen multiple-choice questions comprised the knowledge assessment (i.e., “When treating a patient in the jail ward, is a custody officer/sheriff required to stay in the room with you?). Higher scores indicated more knowledge. The survey concluded with the assessment of attitude by asking for 5-point Likert scale responses to 10 statements about justice-involved health (i.e., “If I were to work in a prison or jail ward, I would feel unsafe” and “I am comfortable talking to a patient about their experiences in jail or prison”). Lower attitude scores indicated less self-identified preparedness and sympathy towards justice-involved populations. The complete survey is available in Appendix A.

### 2.4. Data Analysis

We evaluated differences in demographic and baseline characteristics between the four classes of medical school standing (years 1–4) and tested for differences using ANOVA for continuous variables and Fisher’s Exact Test for categorical variables. Post hoc, Bonferroni-adjusted pairwise comparisons were subsequently conducted when *p* < 0.05 for the ANOVA omnibus test.

To evaluate the total attitude and knowledge scores, we used linear regression, relying on the Central Limit Theorem to assume a normal distribution. A priori covariates of interest included race, age, self-identified gender (male, female, or other), personal connection with the justice-involved setting (e.g., friend or family member has been incarcerated), time working in the jail ward (hours), and amount of education received (hours). We included in our final model only those variables that were statistically significantly associated with the total score(s) or those that show evidence of confounding the relationship between standing and score(s) (i.e., gender and race). We defined a confounder as those variables that alter the effect (β) by >15%. To test model fit, we used the Hosmer-Lemeshow goodness of fit test and conducted an inspection of outliers and influential points. Here, we report β as the effect size. β can be interpreted as the change in experiential or knowledge scores for each one-unit change in the covariate. All analyses were conducted using Stata 15.0 (Statacorp, College Station, TX, USA).

We conducted a post hoc power analysis to determine the effect sizes we would be able to detect for differences in experience and knowledge between groups. A total of 295 surveys were available for analysis. Assuming a two-sided Type I error rate of 5% and 80% power, we will have the ability to detect a moderate difference in effect size of d = 0.41 (Cohen’s d). Power was calculated based on a multivariable linear regression with four independent variables. We used PASS 14.0 (NCSS, LLC., Kaysville, UT, USA) to calculate the effect size we could detect.

## 3. Results

752 email invitations were sent to all current medical students to participate in the survey. In total, 352 surveys were started, and 295 responses were completed and analyzed (response rate of 39.2%) for a final sample size of 295 participants. Response rate calculation was based on all non-respondents being eligible because the survey was sent to specifically categorized students who met eligibility requirements. Partially completed surveys were excluded from response rate.

In total, 74, 86, 62, and 73 responses were recorded from MSIs, MSIIs, MSIIIs, and MSIVs, respectively; 42.7% (126/295) of responses were from self-identified males, 55.9% (165/295) were from self-identified females, and 1.4% (4/295) were from self-identified “other”, which included transgender male, transgender female, gender variant/nonconforming, and “other.” (Table 1).

There were no statistically significant differences between groups with respect to gender or whether the participant had a prior graduate degree (*p* > 0.05 for both). There was a statistically significant difference in age across standing, with those in the later years of medical school being approximately one year older than the prior year (*p* < 0.01). Overall, participants reported highly variable training and experience over the course of medical school. Specifically, participants had statistically significant differences in the amount of education received, hours worked in a jail setting, substance abuse training, and whether they had ever volunteered (*p* < 0.01 for all). In a carceral healthcare setting, 89.2% of first-year medical students and 7.1% of fourth year medical students reported working zero hours. There was a consistent increase in overall knowledge score across the four groups, with the highest knowledge score among the fourth-year students (*p*_trend_ < 0.01). Similarly, the attitude score also increased by year, with the highest score again among the fourth-year students (*p*_trend_ < 0.01).

When assessing the relationship between the overall knowledge score and educational and experiential exposures, we find that knowledge increased with exposure (Table 2). More specifically, an increase in overall knowledge score was observed as carceral medicine education increased (*p*_trend_ = 0.02). The same was true for increasing hours working in a jail (*p*_trend_ < 0.01) and increased substance abuse training (*p*_trend_ < 0.01).

Overall attitude score increased with the number of hours working in a jail (*p*_trend_ < 0.01) and the amount of substance abuse training (*p*_trend_ < 0.01). While any amount of prison education improved attitude score over no education, the improvement was statistically significantly greater in the group who received 1–6 h of training compared to those who received more than 6 h (*p*_trend_ < 0.01).

Last, we found a significant trend of increasing knowledge and attitude scores as the year of standing in school increased (*p*_trend_ < 0.01 for both, Table 3).

## 4. Discussion

We found that, by their fourth year, most students had worked in a carceral health setting. Further, both didactic and experiential learning correlated with medical students’ improved knowledge of and attitude toward justice-involved patients, with both knowledge and attitude scores increasing as the year in medical school increased.

We found that most (89.2%) first-year medical students reported working 0 h in a carceral healthcare setting, compared to just 7.1% of fourth year medical students who reported no work with justice-involved patients. This demonstrates that, despite no formal justice-involved healthcare curriculum, USC medical students are exposed to justice-involved patients over the course of medical school and likely gain knowledge and familiarity with carceral populations over the course of their training. This finding underscores the importance of equipping students in the pre-clinical years with foundational skills and education to support them as they interact with justice-involved patients in the clinical years of school.

Not surprisingly, however, students who worked more hours in a jail or prison setting and who received more substance abuse training scored higher on both metrics. This suggests working with the justice-involved population as a medical student increases knowledge, preparedness, and sympathy towards justice-involved patients. We postulate that enhanced competence could encourage medical students to choose careers in the correctional system and improve care delivered to justice-involved patients. This was reinforced in a 2018 paper which used Social Cognitive Career Choice Theory (SCCT) to demonstrate that placement in carceral settings and improvements in self-confidence and efficacy can enhance interest in working with justice-involved patients [3]. Additionally, students who had more than 6 h of substance use disorder training scored higher on both metrics. This reinforces the notion that the justice-involved population and the substance use disorder population are closely linked [13]—in our study, experience in or knowledge of substance use disorders is associated with both more favorable attitudes and more factual knowledge of carceral health.

Students farther along in their education were able to answer the knowledge questions with higher accuracy. The statistically significant trend of increasing effect on scores with ascension through medical school suggests improving knowledge with increased exposure to justice-involved populations. We hypothesize this is due to the increasing care delivery to this population as they spend more time in the clinical environment at LAC + USC. The βs for knowledge increased over each year in school, indicating increasing levels of knowledge as standing increased. βs representing favorable attitudes followed a similar trend (−0.3, 1.4, and 2.7, respectively) compared to first year students. Although we were not able to find existing studies that explored the evolution of medical student attitudes and knowledge towards justice-involved populations, one study found increasingly negative attitudes towards underserved patients over the four years of medical school—the opposite of our findings [15]. Another study of medical student attitudes towards mental illness found that there was no change in attitudes with more exposure to psychiatric patients [16]. This may suggest that the evolution of medical student attitudes with increased exposure may not be uniform across all stigmatized populations. It is important to acknowledge that many variables, such as differences in study objectives and design, rather than actual population differences, may have led to these observations. Our results, however, suggest that exposure and attitudes toward justice-involved patients may be intertwined. The observed trends indicate that there was an increase in both knowledge and favorable attitudes towards the justice-involved population with more time working in the carceral setting.

Despite demonstrated improvement across the years of medical school, knowledge scores were still low, suggesting that experiential learning is not sufficient. An effective formal curriculum, paired with clinical experience, could be beneficial to supplement or, ideally, replace incomplete and inconsistent teaching in the clinical setting. Furthermore, early intervention in the pre-clinical years could better prepare students to enter their clinical years already equipped with the knowledge and attitudes to serve the justice-involved population. Integrated justice-involved health curricula are an important step toward better educating and training our healthcare workforce in health equity for carceral populations.

This was a single-center study that may not be generalizable to all justice-involved practice settings at all US medical schools, especially due to the unique relationship and intercalation of care for patients from the Twin Towers Correctional Facility and the LAC + USC Medical Center. Correctional policies vary county by county, and answers to the didactic portion of the survey may not be generalizable on a national level. Additionally, a single instrument, which was significantly modified from the Substance Abuse Attitudes Survey was used. Although it was evaluated and approved by medical education experts at our institution, the survey would ideally be pilot tested on a larger group of respondents and validated by formal survey development guidelines [17]. To ensure diversity of answers and generalizability of data in the future, more than one instrument should be used, and the authors suggest adding the Attitudes Towards Prisoners instrument, first validated in 1985 and used in many studies since then, as an adjunct measure [18,19]. Another limitation is the potential for self-selection bias. Because only 39% of those invited to participate did so, it is possible that those who were more interested in the topic, for any variety of reasons, were more likely to participate. The effect of selection bias is difficult to assess in this scenario as we are not able to compare informal exposure to justice-involved patients between responders and non-responders.

## 5. Conclusions

In this single-institution survey of medical students’ experiences with justice-involved patients, most fourth-year students had worked in a carceral health setting. Didactic and experiential learning opportunities correlated with improved knowledge of and attitude toward justice-involved patients, with increases in both knowledge and attitude scores increasing as the year in medical school increased. Despite improvement, senior medical students still scored poorly on the knowledge portion. These findings underscore the need for a formal curriculum to improve medical student knowledge and attitudes.

## Figures and Tables

**Table 1 healthcare-09-01302-t001:** Demographics of Study Population.

Total Responses	295	74	86	62	73	
**Age (Mean ± SD)**	**25.2 (0.1)**	**24.3 (0.2)**	**24.5 (0.2)**	**25.8 (0.3)**	**26.5 (0.1)**	**<0.01**
Gender						
Male	126	31 (24.6)	31 (24.6)	32 (25.4)	32 (25.4)	0.34
Female	165	43 (26.1)	53 (32.1)	30 (18.2)	39 (23.6)	
Other	4	0	2 (50.0)	0	2 (50.0)	
Race						
White	85	26 (30.6)	27 (31.8)	14 (16.5)	18 (21.2)	Not calculable
Asian	123	30 (24.5)	35 (28.5)	28 (22.8)	30 (24.4)	
Black or AA	18	4 (22.2)	4 (22.2)	4 (22.2)	6 (33.3)	
Latino or Hispanic	38	9 (23.7)	13 (32.2)	4 (10.5)	12 (31.6)	
Mixed Race	20	5 (25.0)	7 (35.0)	5 (25.0)	3 (15.0)	
Other	5	0	0	4 (80.0)	1 (20.0)	
Prefer not to answer	6	0	0	3 (50.0)	3 (50.0)	
Prior Graduate Degree						
None	264	66 (25.0)	77 (29.2)	54 (20.5)	67 (25.4)	0.72
MPH	4	1 (25.0)	2 (50.0)	1 (25.0)	0	
PhD	1	0	0	1 (100)	0	
MS	19	3 (15.8)	6 (31.6)	5 (26.3)	5 (26.3)	
Other	7	4 (57.1)	1 (14.3)	1 (14.3)	1 (14.3)	
Prison Medicine Education						
0 h	224	65 (29.0)	67 (29.9)	38 (17.0)	54 (21.1)	<0.01
1–6 h	61	6 (9.8)	18 (29.5)	21 (34.4)	16 (26.2)	
More than 6 h	10	3 (30.0)	1 (10.0)	3 (30.0)	3 (30.0)	
Hours worked in Jail						
0 h	156	66 (42.3)	69 (44.2)	16 (10.3)	5 (3.21)	Not Calculable
1–6 h	72	8 (11.1)	16 (22.2)	32 (44.4)	16 (22.2)	
More than 6 h	67	0	1 (1.49)	14 (20.9)	52 (77.6)	
Hours Substance Use Training						
0 h	65	40 (61.5)	10 (17.0)	11 (16.9)	4 (6.15)	<0.01
1–6 h	183	30 (16.4)	63 (34.4)	38 (20.8)	52 (28.4)	
More than 6 h	47	4 (8.5)	13 (27.7)	13 (27.7)	17 (36.2)	
Volunteered in Carceral Setting						
No	261	59 (22.6)	75 (28.7)	55 (21.2)	72 (27.6)	<0.01
Yes	33	15 (45.5)	11 (33.3)	6 (18.2)	1 (3.03)	
Overall Knowledge Score		7.6 (0.2)	7.9 (0.2)	8.8 (0.2)	9.2 (0.2)	<0.01
Overall attitude score		32.3 (0.5)	31.8 (0.6)	34.9 (0.7)	37.2 (0.7)	<0.01

Footnotes: categorical variables reported as count (%) and *p*-value assessed using Fischer’s exact chi-square test. Continuous variables reported as mean/median (SE) and *p*-value assessed using ANOVA test.

**Table 2 healthcare-09-01302-t002:** Univariate Associations between Didactic and Experiential learning and assessment score.

	Percent Score	Mean ± SD	Beta (95% CI)	*p*-Value	*p* _trend_
Overall Knowledge Score					
Formal Carceral Medicine Education					0.03
0 h	54.7%	8.2 ± 2.0	Ref		
1–6 h	58%	8.7 ± 1.9	0.5 (−0.1, 1.0)	0.07	
More than 6 h	60.7%	9.1 ± 1.2	0.9 (−0.3, 2.1)	0.14	
Hours Worked in Jail					<0.01
0 h	53.3%	8.0 ± 2.0	Ref		
1–6 h	56.0%	8.4 ± 1.6	0.4 (−0.1, 1.0)	0.11	
More than 6 h	60.7%	9.1 ± 1.8	1.1 (0.6, 1.7)	<0.01	
Substance abuse Training					<0.01
0 h	53.3%	8.0 ± 2.1	Ref		
1–6 h	55.0%	8.2 ± 1.9	0.3 (−0.2, 0.9)	0.24	
More than 6 h	61.0%	9.1 ± 1.7	1.3 (0.6, 2.0)	<0.01	
Overall Attitude Score					
Formal Carceral Medicine Education					<0.01
0 h		32.9 ± 5.6	Ref		
1–6 h		37.2 ± 5.0	4.3 (2.8, 5.9)	<0.01	
More than 6 h		37.0 ± 5.7	3.8 (0.4, 7.3)	0.03	
Hours Worked in Jail					<0.01
0 h		32.3 ± 5.2	Ref		
1–6 h		33.9 ± 5.4	1.7 (0.2, 3.2)	0.02	
More than 6 h		37.9 ± 5.3	5.7 (4.2, 7.2)	<0.01	
Substance abuse Training					
0 h		31.7 ± 4.9	Ref		<0.01
1–6 h		33.5 ± 5.4	1.8 (0.3, 3.3)	0.02	
More than 6 h		38.3 ± 5.8	6.5 (4.5, 8.5)	<0.01	

Reported means (SE) and *p*-value assessed using multiple linear regression adjusted for gender and race, Beta is interpreted as the increase in mean outcome (overall knowledge score or overall Likert score) per unit increase in exposure.

**Table 3 healthcare-09-01302-t003:** Associations between knowledge score, attitude score and standing year in Medical score.

	Mean ± SD	Beta	*p*-Value	*p* _trend _
Overall Knowledge Score				<0.01
MS1	7.6 ± 2.0	Ref		
MS2	7.9 ± 2.0	0.4 (−0.2, 0.9)	0.21	
MS3	8.7 ± 1.3	1.2 (0.5, 1.8)	<0.01	
MS4	9.2 ± 1.9	1.6 (1.0, 2.2)	<0.01	
Overall Attitude Score				<0.01
MS1	32.3 ± 4.5	Ref		
MS2	31.8 ± 5.3	−0.4 (−2.1, 1.2)	0.62	
MS3	35.0 ± 5.9	2.7 (0.8, 4.6)	<0.01	
MS4	37.2 ± 5.6	5.0 (3.2, 6.7)	<0.01	

Reported means (SE) and *p*-value assessed using multiple linear regression adjusted for gender and race. Beta is interpreted as the increase in mean outcome (overall knowledge score or overall Likert score) per unit increase in exposure.

## Data Availability

The data presented in this study are available on request from the corresponding author. The data are not publicly available to protect the privacy of respondents.

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
