# Peer review of "Medical Students’ Knowledge and Attitudes Regarding Justice-Involved Health"

_healthcare, 2021, doi:10.3390/healthcare9101302_

Round 1

Reviewer 1 Report

This is an important study in an area which has had little research published as identified by the authors. The  focus on knowledge and attitudes is logical and highlights the value of such an education program.  There are a number of references to a Carceral Health Curriculum the authors could elaborate on detail of the core components of such a program.  The survey is modelled on Substance Abuse Attitudes Survey, it was not clear if it had significant revision to include a broader focus for this population. In lines 230-235 it was noted some studies had not shown similar effects. There are many variables which could lead to this difference. A further description of differences of those interventions and study objectives may be of assistance. In understanding this. Overall excellent piece of research

Reviewer 2 Report

This research shows the need for a formal curriculum to enhance the knowledge of medical students and attitudes regarding Justice Involved Health.

The paper is clearly written and should be understood by researchers.

My comments are:

- Please change the keywords using MeSH main heading terms (Medical Subject Headings 2020)

- Low number of references. A literature review is recommended to include more studies or justify the lack of research in the area of study.

- I consider it necessary to indicate the calculation of the sample size.

- Description of questionnaire validation data.

Round 2

Reviewer 2 Report

The authors have responded adequately to all indications. I have no further suggestions.